# A Preliminary Study of the Characteristics of Radon Data from Indoor Environments and Building Materials in the Campania Region Using PCA and K-Means Statistical Analyses

Carlo Sabbarese [1,*], Maria Luisa Feola [1], Fabrizio Ambrosino [2], Vincenzo Roca [1], Antonio D'Onofrio [1], Giuseppe La Verde [2], Vittoria D'Avino [2], Mariagabriella Pugliese [2] and Vittorio Festa [3]

[1] Department of Mathematics and Physics, University of Campania "L. Vanvitelli", Viale Lincoln 5, 81100 Caserta, Italy; marialuisa.feola@studenti.unicampania.it (M.L.F.); vroca222@gmail.com (V.R.); antonio.donofrio@unicampania.it (A.D.)
[2] Department of Physics "E. Pancini", University of Naples Federico II, Via Cinthia ed. 6, 80126 Naples, Italy; fabrizio.ambrosino@unina.it (F.A.); glaverde@na.infn.it (G.L.V.); vdavino@na.infn.it (V.D.); mpuglies@na.infn.it (M.P.)
[3] Protection Solutions S.r.l., 80133 Naples, Italy; festav@yahoo.com
* Correspondence: carlo.sabbarese@unicampania.it

**Abstract:** For a healthy indoor environment, it is important to understand which materials and factors favor the generation of high levels of indoor radon. A preliminary multivariate statistical analysis was carried out on two datasets concerning indoor radon and building materials in the Campania Region using Principal Component Analysis (PCA) and the k-means partitional analysis technique. A total of 13 parameters related to building materials were used. The results show the greater contribution of building materials of volcanic origin to the concentration of indoor radon and thoron activity and the different influence of the parameters of the 31 materials analyzed. The same analyses applied to the indoor radon values of 694 rooms in the Campania Region were equally effective in assessing the structural characteristics of indoor environments that most influence indoor radon levels. The study provided an effective assessment of the influence on radon activity of several environmental parameters, which are often not adequately considered.

**Keywords:** indoor radon; Campania Region; PCA; k-means; building materials

## 1. Introduction

Radon ($^{222}$Rn) is a radioactive natural gas that comes from the decay of radium ($^{226}$Ra) present in soils and building materials and tends to accumulate in indoor environments [1,2], where in some cases it can reach very high concentrations to represent a significant risk for human health because it can generate a carcinogenic process [3,4].

It is therefore important to understand which materials and factors favor the generation of high levels of indoor radon [5–7]. The analysis and evaluation of the different characteristics of indoor environments, in different types of buildings, in different environmental contexts and linked to different types of use, is a complex problem that requires careful evaluation of situations that are often poorly documented and not easily quantifiable because many agents contribute through multiple dynamics to the determination of the radon activity concentration [8–10].

The numerical solutions of the related transport equations are solved and used to estimate the activity concentration of radon in closed environments e.g., [11]. These investigations lead to the conclusion that, in addition to the nature of the soil and building materials, the location of the lower floor of houses plays a significant role in determining the amount of radon entry into residential buildings.

The Campania Region (southern Italy) is characterized by soils and materials used for construction of volcanic origin, and therefore richer in natural radionuclides [12]. Measurements of all possible parameters characterizing the generation and emission of radon and thoron ($^{220}$Rn) of 31 materials have been performed [13,14]. Indoor measurement projects have been carried out [15–23] and a first potential indoor radon map of the Campania Region was also drawn up [24]. During a campaign, the values of various parameters that are considered to influence the accumulation of indoor radon are also generally acquired but are often not adequately used to implement analysis procedures. The availability of this large dataset makes it possible to perform a more detailed analysis to distinguish and characterize the different sources and parameters influencing the indoor radon levels. Here, multivariate statistical tools are applied to derive data characteristics on both indoor radon and materials used in construction [24].

## 2. Materials and Methods

### 2.1. Data

The first dataset used here consists of the results of the work to characterize the content of natural radioactivity in 31 natural building materials typical of the Campania Region using high-resolution spectrometric methods [13]. The analyzed samples of natural building materials from the Campania Region were selected to be representative of different geological environments of the region. A total of 31 samples were collected: 8 tuffs (brown, grey, yellow, green), 2 pumices, 2 lava stones, 2 marbles, 3 limestones, 1 bauxite, 2 pozzolans, 2 ignimbrites, 2 tephrites, 2 siltstones, 4 sandstones and 1 clay. Several samples of a given building material belong to different geographical locations with the same geological settings. The data for each material concern the activity concentration of the radon and thoron progenitors, their emanation coefficient, the exhalation rate, and risk indices such as the activity concentration I index, the $H_{ex}$ and $H_{in}$ indices for assessing external and internal exposure to gamma radiation [13].

The second dataset consists of a subset of the radon indoor data used for the realization of the first potential map of radon distribution $^{222}$Rn in the Campania Region. This map is based on a geostatistical interpolation method kriging through 10 proxy variables used as predictors influencing the gas emission [24]. The radon indoor activity concentration, expressed in Bq/m$^3$, was obtained as an annual average of two half-yearly measurements performed using SSNTD methods. Each radon data point is accompanied by different data characterizing the measurement environment and the building in which it is located.

### 2.2. Statistical Analysis Methods

#### 2.2.1. PCA

Principal Component Analysis (PCA) is particularly suitable for studying the structure of a set of multivariate observations, especially when there is no prior information on how the measured variables are dependent or associated with each other; it is a multivariate statistical methodology that allows many variables to be analyzed simultaneously, guaranteeing synthesis, ease of reading and the least possible loss of information [25–27]. It replaces the p variables of a data matrix with a new set of artificial variables called principal components that are uncorrelated and listed in descending order of their variance. These new variables are linear combinations of the observed variables, so it is possible to obtain as many variables as the original ones; the choice of the number of components considered suitable for interpretation follows certain criteria. Therefore, the objective of the analysis is to determine the axes that allow the best representation of the point cloud, preserving the original variability and minimizing the loss of information [27].

As a first step we construct a rectangular unit-variable table of dimension (n, p) that we will denote Y, in which n are the units and p the quantitative variables revealed to then obtain a representation of the n units in the space $R^p$ generated by the p variables and a representation of the p variables in the space $R^n$ generated by the n units.

In the space of the variables, a base formed by orthogonal vectors of unitary norm is determined. It is obtained as linear combinations of the original variables and is able to represent at best the structural information of the system with respect to a fixed optimization criterion. This criterion must be chosen in such a way as to guarantee the search for axes that maximize the sum of the squares of the distances between all the possible pairs of point-units projected onto them. The best axis, the best plane, or the best subspace on which to project the point-units is then searched for, making sure that the original distances between all pairs of points are represented in the projection with the least possible distortion.

The PCA performs algebraic transformations of the initial matrix Y that result in a translation of the point system at its barycenter, thus generating a new centered data matrix X. Applying a normalization condition on the principal axes and the Lagrange multiplier formula, the problem is a maximization with a normalization constraint: we look for the maximum value eigenvalues of the matrix X'X, a full-rank matrix p with distinct eigenvalues which will correspond to the correlation matrix. The eigenvalues precisely represent the amount of the total variability observed in the original variables, expressed by each principal component. The eigenvectors corresponding to the chosen eigenvalues identify the new orthogonal axes. In the space of individuals $R^n$, the matrix X defines a cloud of p points, and the search for factors that maximize the variance of the projected points is analogous to that in the space $R^p$. The construction of the plans on which to project the points is carried out by looking for the "significant" factors, that is, those factors that consider the percentage of variability explained and are easy to interpret. The scree test is a criterion that makes it possible to plot the eigenvalues on a histogram and identify abrupt changes in the values of the sample eigenvalues. The visualization of the results takes place with a circle of unit radius drawn from the matrix of correlations between variables and components. Each variable is represented on this circle as a vector, whose extreme point coordinates are the correlations between the new variables and the original ones. The lower the angle of the representative vector joining the origin of the circle to the point of the variable, the higher the correlation of the variable to the component represented by the axis.

For individuals, a factor map is constructed similar to a Cartesian plane with the aim of analyzing the behavior of individuals in relation to the components according to the position they occupy on the plane itself [27].

### 2.2.2. K-Means

Partitive methods are automatic classification methods with the objective of iteratively defining the classification structure of a set under study [28]. The dataset is subdivided into K non-empty subgroups, where K is the number of clusters assigned as input to the algorithms that will stop as soon as the subdivision appears stable with respect to the evaluation criterion used, guaranteeing that each instance is placed exactly in one of the many mutually exclusive clusters generated. The generated clusters are usually exhaustive and exclusive in the sense that they cover all the variability of the set, and each observation belongs to one and only one cluster [28].

The different quality measures used tend to express the degree of homogeneity of observations belonging to the same cluster and their heterogeneity with respect to observations in other clusters. Partition methods are, therefore, heuristic in nature and operate at each step the choice that appears locally more advantageous; the result is very much conditioned by the initial choice. For this reason, the algorithm is run several times with different initial choices, remembering that clustering is good if there is a small percentage of variability in clusters and a large percentage of variability between clusters.

To identify the optimal number K of clusters that guarantees the best solution, there are several criteria, one of which is based on the calculation of the silhouette index. The silhouette index is given by the ratio of the divergence of the average distance between a generic object and the other points of the nearest of the other classes and the average distance between the same object and the other points within its same class, on the maximum

of the two distances. The closer the index is to unity, the better the classification result obtained [28].

The algorithm chosen for the classification operation conducted in this work is the K-means algorithm. This algorithm divides the dataset into K non-empty subgroups, where K is the number of clusters assigned as input. Randomly choosing among all the observations, a number K of them that are the centers of the generated clusters. The new centers are defined iteratively starting from the various observations already clustered and every other observation is placed in the cluster with the closest center to it, and therefore, a definition of distance must be used to evaluate the proximity between objects; the most commonly used distance is the Euclidean distance of the sum of the squares of the deviations.

### 2.2.3. R Software and RStudio

Both the analyses and the classifications of the datasets were carried out using the RStudio software, GUI (Graphical User Interface) for the R programming language [28]. This language is interpreted, open source and object-oriented [29].

RStudio is an integrated development environment (IDE) for R, which can be managed through the RStudio console [30]. It is, therefore, a statistical package, a set of macros, libraries and objects that can be used to manage and analyze data and produce graphs. It has several libraries and packages.

The functions *PCA()* for Principal Component Analysis and *k-means()* for automatic clustering are contained in the packages FactoMineR and Stats, respectively. The first function provides a list of elements such as a matrix containing all the calculated eigenvalues with their respective percentages of variance and cumulative variance, matrices relating to variables and individuals containing information such as coordinates, contributions and correlations, and matrices containing information relating to any qualitative variables. The second function provides a list of data relating to clusters, the number of which must be provided as input [29].

## 3. Results and Discussion

### 3.1. Building Materials

In the first step of applying the PCA analysis, we determined the eigenvalues of the correlation matrix, the percentage of explained variability and the percentage of cumulative variability for each principal component to choose the appropriate number of dimensions to consider. The trend in the eigenvalues as a function of the number of principal components (scree plot in Figure 1) is decreasing. The point at which an abrupt change in slope occurs identifies the number of components to be considered. In this case, the first two dimensions have a high percentage of explained variability, the sum of which is about 93%, which is excellent for proceeding with the analysis. Moreover, even the criterion of eigenvalues greater than 1 would lead us to choose the first and second dimension: Dim.1 = 8.101, Dim.2 = 3.968.

The eigenvalues chosen are, therefore, the variance of component 1 and component 2, respectively.

The correlation circle obtained from the PCA is in Figure 2a. The emanation coefficients of $^{222}$Rn and $^{220}$Rn are all located in the first quadrant, and are thus highly correlated with dimension 2 and weakly correlated with dimension 1, whereas the exhalation rates, both per unit area and per unit mass, are averagely correlated with both dimensions. The risk indices are all placed in the fourth quadrant, together with the activity concentration of $^{226}$Ra, $^{232}$Th and $^{40}$K, indicating they are correlated with dimension 2 and anticorrelated with dimension 1.

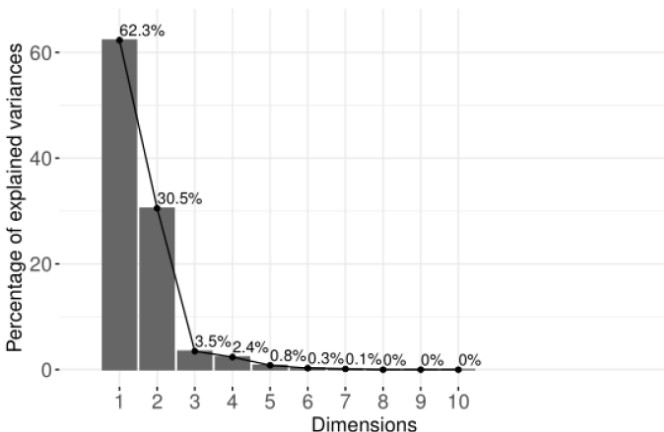

**Figure 1.** Scree plot of the eigenvalues of the factors with respect to the principal components obtained from the building materials dataset.

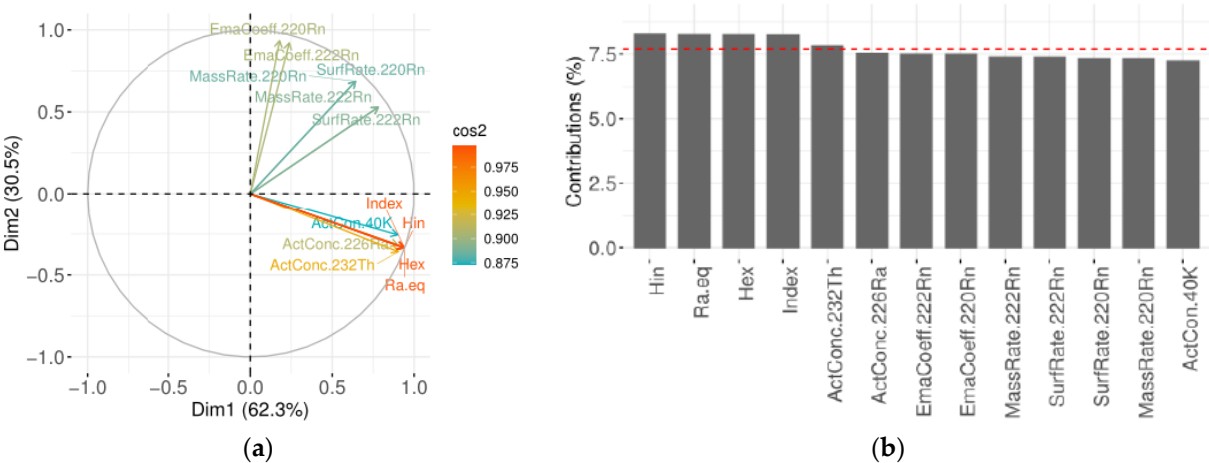

(**a**)           (**b**)

**Figure 2.** (**a**) The correlation circle obtained from the PCA; the palette of colors combined with the different values of cos$^2$ allows a faster interpretation of the results. (**b**) Percentage contribution of each variable to the determination of the first and second component.

Figure 2b shows the contribution of each variable in the determination of the factorial plans; the red dashed line corresponds to the cut-off value, i.e., the average expected percentage of a variable's contribution (about 8%). The first four variables reported, i.e., the hazard indices, are therefore, those that contributed most to the definition of the main components.

After completing the analysis in the space of variables, we analyze the space of n individuals by repeating all the steps performed in the case of the variables. A multidimensional study in this space is based on the choice of a metric and a diagonal matrix D to define the inertia of the points. Inertia is a value that measures the dispersion of the point-individual cloud N(I) and is equivalent to the sum of the weights of everyone by the squares of the unit vector norms. The metric, which is encompassed by a matrix M, is used to calculate the norm of each squared vector, whereas the matrix D is the diagonal matrix of the weights of the n units. These data in the study are standardized and centered, so M = I. There is no reweighting because the data are homogeneous, and the metric is Euclidean.

The objective at this stage of the analysis is to maximize the centered value of the data with respect to the origin and thus maximize inertia. We first generate the factorial map of individuals to analyze their behavior.

Figure 3a shows the representation of all the materials in the plane of the two main components identified. Volcanic materials such as tuff, pumice stone and pozzolan are separated from the others. Yellow tuff is very close to dimension 2 and is positively

correlated with it, demonstrating a high rate of radon emission and exhalation. Pumice also occupies a position that demonstrates its high radioactivity content, and compared with yellow tuff, it has a higher risk index. Pozzolan and Lavic Stone are positively correlated with dimension 1; Pozzolan (2) and Lavic Stone (2) occupy an extreme position, which suggests that these materials are of high radiological risk. Carbonate, siliciclastic and sedimentary rocks such as siltstone, limestone, marble, sandstone are positioned close to the origin of the axes and are negatively correlated with the two dimensions, especially with the dimension related to radioactive risk. In fact, these types of materials have lower levels of radioactivity and low capacity to contaminate the external environment, making them approved materials for building construction.

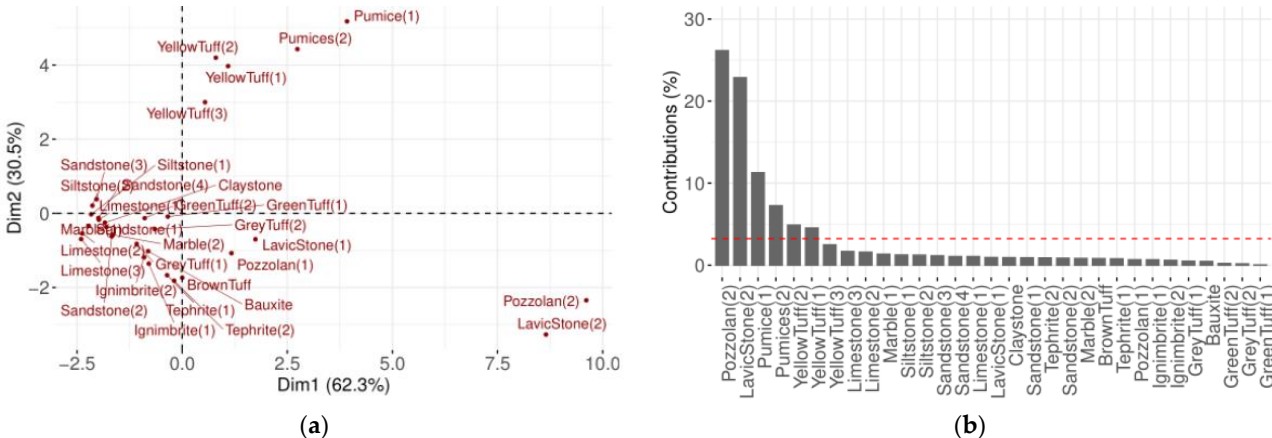

(a)                                    (b)

**Figure 3.** (**a**) Representation of all Campania building materials in the plan of the two main components identified. (**b**) The percentage contribution of each material (individual) to the two main components.

The cut-off percentage of the contributions of the individuals (Figure 3b) is lower than that calculated in the case of the variables because the relationship is formed with a higher number of individuals (31). The individuals Pozzolan (2), LavicStone (2), Pomices (1), Pomices (2), Yel-lowTuff (2), YellowTuff (1) are those that contributed most to the determination of the components.

The biplot of Figure 4a, a superimposition of the factorial plane and the correlation circle, shows the positioning of the individuals based on the two main components identified.

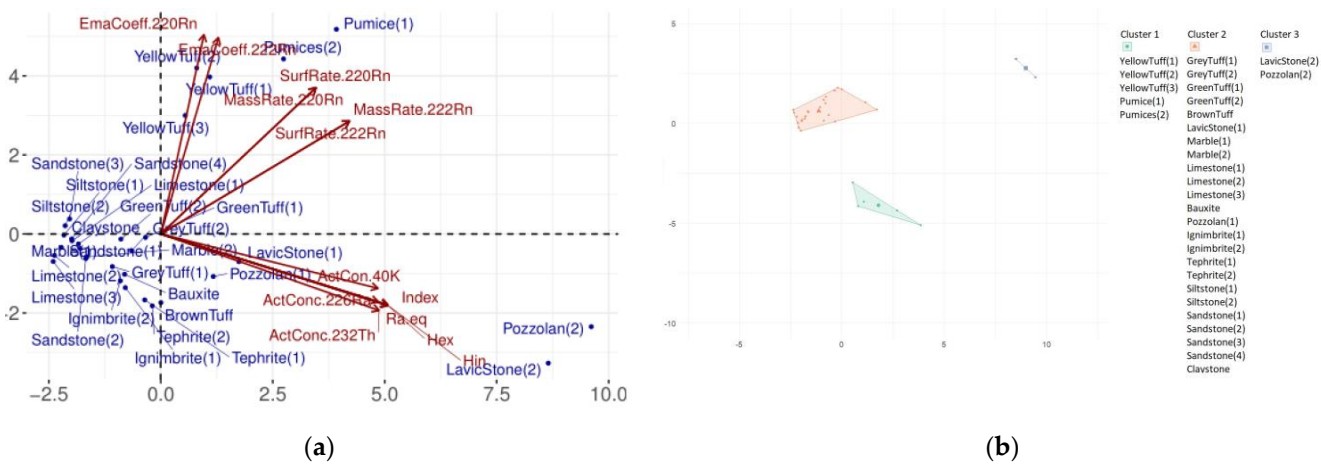

(a)                                    (b)

**Figure 4.** (**a**) The plot of Figure 3a combined with the representation of the parameters to highlight the influence of each on the materials. (**b**) Representation of the three clusters into which the materials were divided by the k-means partition analysis based on their similarity.

We now provide a classification of the analyzed dataset by means of a partitional analysis with the K-means algorithm and using the criterion of internal variability. The optimal number of clusters for this dataset is K = 3. The result is shown in Figure 4b. For the sake of brevity, we do not show other plots here that were obtained to highlight the influence of the different parameters on the clustering of the three groups of materials.

### 3.2. Radon Indoor Data

The dataset used for the entire analysis consists of 694 measured indoor radon activity concentration values (expressed in $Bq/m^3$) and the corresponding environment data:

1. the type of building (above ground, underground, basement, on pillars).
2. the floor number.
3. the wall material (concrete, brick, stone, tuff, other).
4. wall covering (paper, plaster, tiles, other).
5. the floor covering (tiles, wood, linoleum, carpet, granite,
6. marble, other).
7. the number of windows.
8. the number of doors.
9. the postcode of the place where the house is located.

To extract more information, several PCAs were carried out by changing the actual and additional variables to choose those with a higher percentage of explained variability. Table 1 lists the four that produced the best results together with the variables used for each, and the first two components whose sum of the percentage value of explained variability is greater than 60% (percentage chosen by means of Kaiser's rule).

**Table 1.** Parameters of the four PCA analyses carried out on indoor radon datasets.

|  | First Analysis | Second Analysis | Third Analysis | Fourth Analysis |
|---|---|---|---|---|
| **Actual variables** | All | Floor number, Building type, Postcode | Floor number, Windows, Wall material, Floor covering, Postcode | Wall material, Wall covering, Floor covering |
| **Additional variables** | None | Wall material, Wall covering, Floor covering, Windows number, Doors number | Building type, Wall covering, Doors number | Building type, Floor number, Windows number, Doors number, Postcode |
| **First component** | Building material, Windows, Floor covering | Floor number | Postcode | Wall covering, Floor covering |
| **Second component** | Postcode, Floor number | Wall material | Building type | Building material, Floor number |

Figure 5a shows the correlation circle of the first PCA analysis. It can be seen that postcode and floor variables are positively correlated with dimension 2, whereas the wall material variable is negatively correlated with it. The variable floor covering is positively correlated, whereas the variable windows is negatively correlated with dimension 1. We can, therefore, interpret dimension 1 as representing the interior environment of the dwelling while dimension 2 represents the environment outside the dwelling. Figure 5b shows that the variable with the greatest contribution to indoor radon is the floor number variable; the postcode and floor covering variables exceed the expected average contribution value of 20% (dashed red line). The percentage of the other two variables is about 15%.

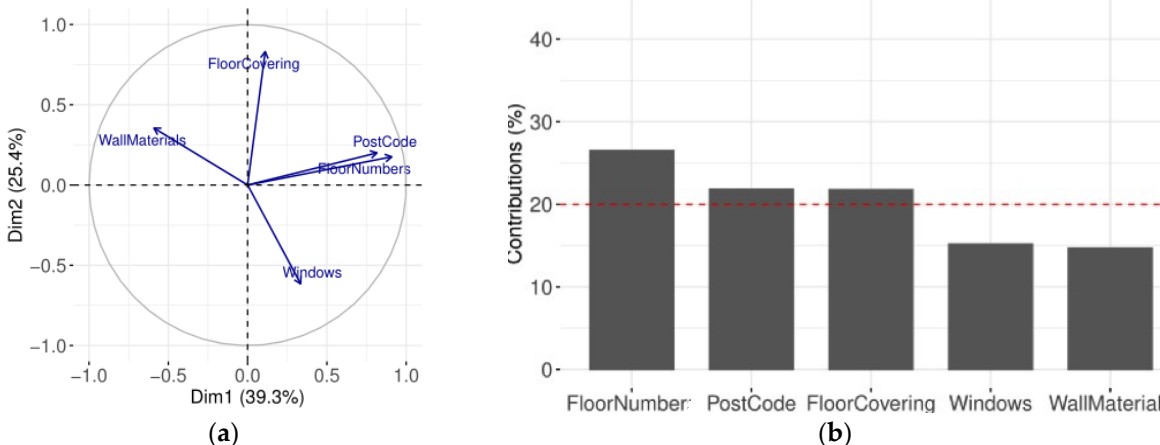

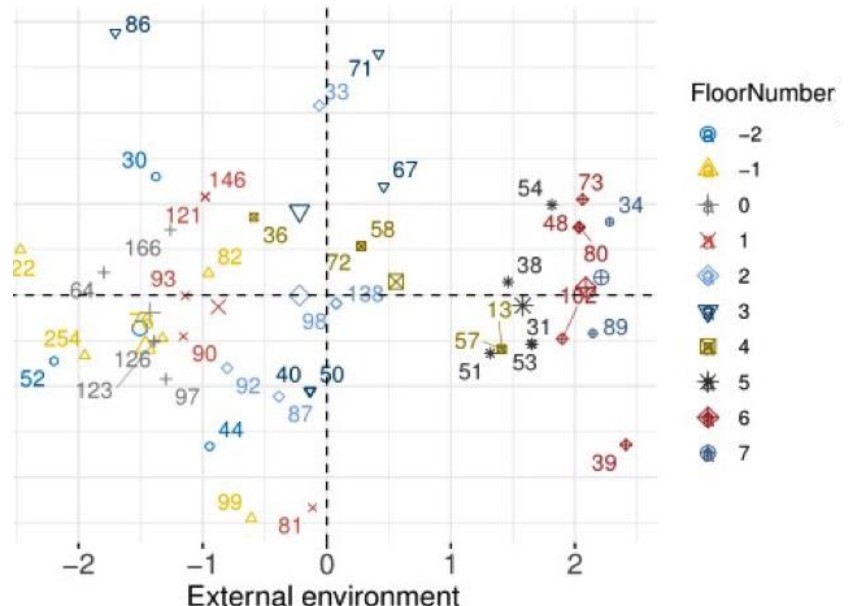

**Figure 5.** Results of the first PCA analysis: (**a**) the correlation circle and (**b**) percentage contributions of the factors that formed the main components (see Table 1).

Figure 6 shows the plot of only a small subset of the data (for reasons of readability of the graph) that distinguishes each indoor radon value according to the floor number of the corresponding dwelling. Most of the indoor radon activity concentration values detected in the 694 dwellings are arranged around the origin of the axes; therefore, they are influenced by both the external environment of the house and the indoor one. However, certain cases can be highlighted; for example, value 86 Bq/m$^3$ is positively correlated in a strong way with the indoor environment dimension (greater influence on the indoor environment), whereas value 40 Bq/m$^3$ is characterized by a positive correlation with the outdoor environment dimension (greater influence on the outdoor environment). With this analysis it is possible to assess the influence on each from the external or internal environment.

**Figure 6.** Factor map of the first analysis with differentiated individuals according to the plan.

The second analysis, carried out with the variables reported in Table 1, shows that the variable floor covering occupies a position close to the axis of dimension 1 and far from that of dimension 2 (Figure 7a); in contrast, the variable wall covering is positioned close to the axis of dimension 2 and far from that of dimension 1. The percentage contributions of the three variables in the identification of the two main components are shown in Figure 7b.

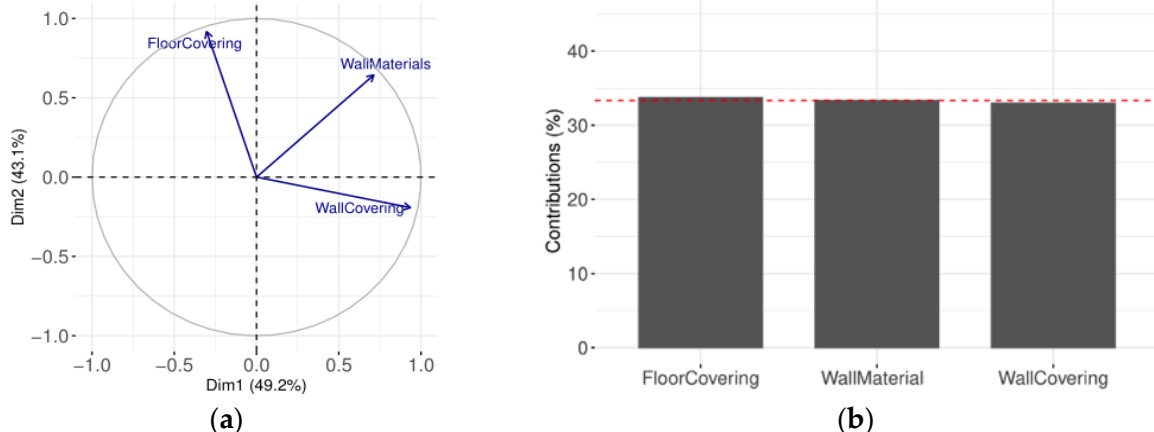

**Figure 7.** Results of the second PCA analysis: (**a**) the correlation circle and (**b**) Percentage contributions of the factors that formed the main components (see Table 1).

Two factor maps were made with a subset of the values: one highlighting the floor covering (Figure 8a) and the other the wall material (Figure 8b). The first graph shows that wood-panelled floors and most linoleum floors are positively correlated with floor size. The corresponding dwellings have an indoor radon value that is more influenced by floor coverings, as is also shown by the position of wood coverings in the fourth quadrant.

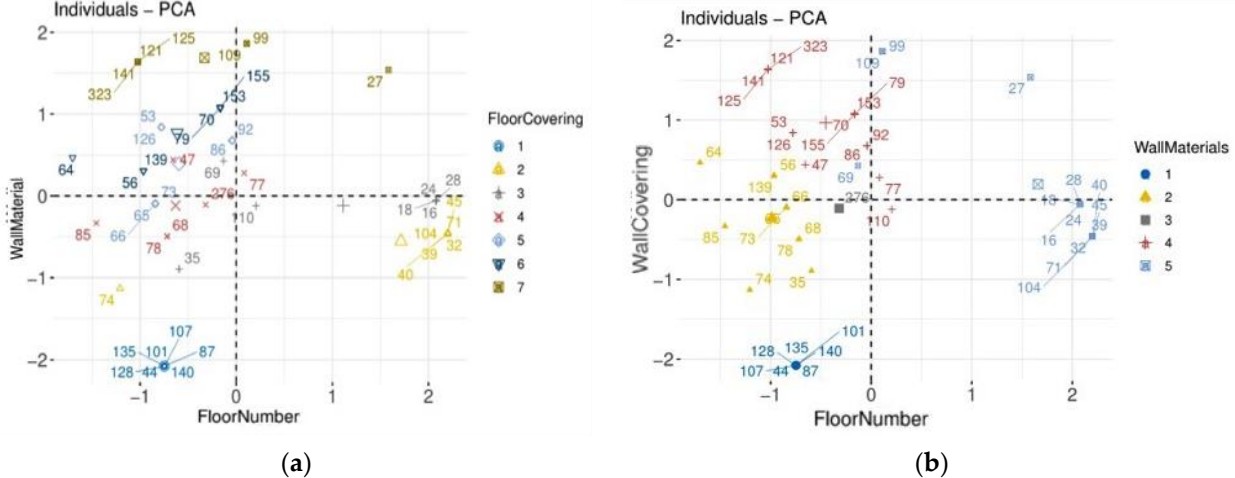

**Figure 8.** Factor maps of the first analysis with differentiated individuals according to (**a**) floor covering (1 = Majolica, 2 = Wood, 3 = Linoleum, 4 = Carpet, 5 = Granite, 6 = Marble, 7 = Other) and (**b**) wall materials (1 = Cement, 2 = Bricks, 3 = Stones, 4 = Tuff, 5 = Other).

Floors covered with tiles are all positioned in the third quadrant, and are negatively correlated with both dimensions. This indicates that the indoor radon value is the result of other factors. Finally, dwellings paved with marble occupy a position that is positively correlated with wall size and negatively correlated with floor size: these dwellings are more influenced by the material and the wall coverings in terms of indoor radioactivity.

Figure 8b shows that dwellings with tuff walls are positively correlated with the wall dimension in contrast to those with concrete walls; dwellings with walls made of bricks are negatively correlated with the floor dimension. The only dwelling with stone walls is located close to the horizontal axis and is negatively correlated with both dimensions. It is evident that the tuff walls in a building have a greater influence on the indoor radon values; similarly, if the walls are made of marble rocks, we should not expect a radon value linked to them but to the choice of floor covering.

The third analysis, carried out with the variables shown in Table 1, shows that the variable floor number is linked to dimension 2, the variable postcode occupies a position closer to dimension 1, and the building type is further away from this dimension (Figure 9a). Dimension 1, therefore, represents the geographical location (position) of the buildings, whereas the second dimension represents the type of building (underground, basement, on pillars), including the floor number and the type of building. The percentage contributions of the three variables in identifying the two main dimensions are shown in Figure 9b. The contribution of the postcode is greater than that of the building type, which in turn is greater than that of the floor number.

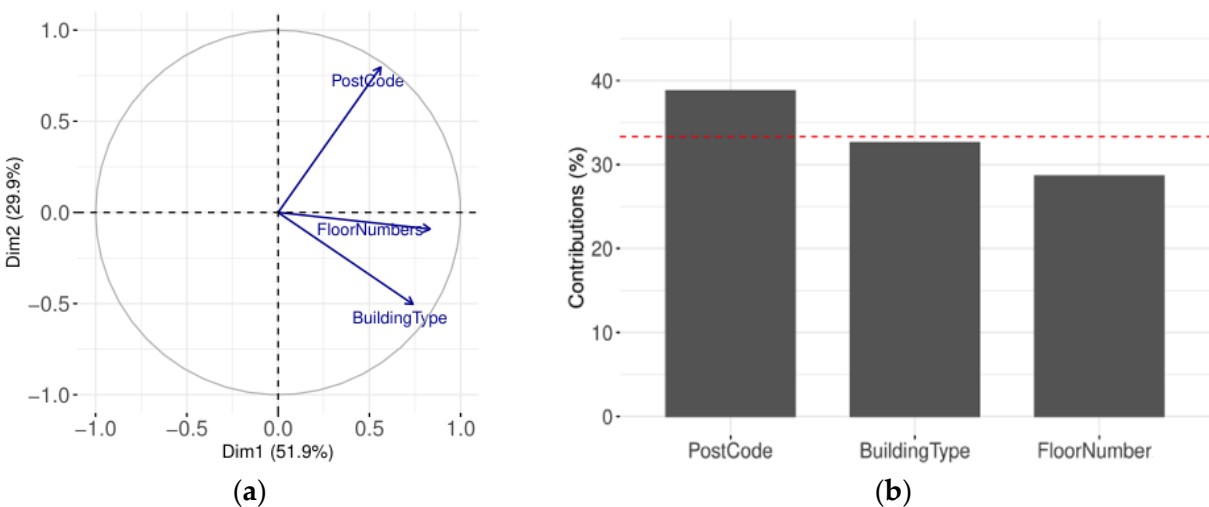

**(a)**                                                                                                              **(b)**

**Figure 9.** Results of the third PCA analysis: (**a**) the correlation circle and (**b**) percentage contributions of the factors that formed the main components (see Table 1).

Figure 10 shows the disposition of a subset of the individuals and their higher correlation with the building structure (e.g., 78, 212, 79 Bq/m$^3$) or with the dimension representing the geographic location (e.g., 126). Additionally, individuals 76 Bq/m$^3$ and 91 Bq/m$^3$ show a position that is slightly different from the average, occupying the fourth quadrant with a negative correlation with the geographic position component but a positive correlation with the building component.

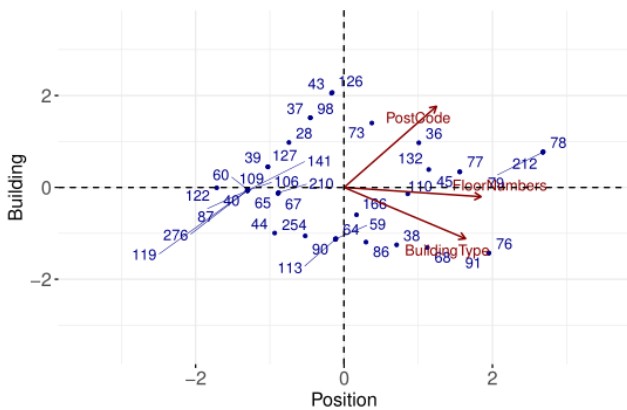

**Figure 10.** Biplot obtained from the third analysis (see Table 1).

The fourth analysis, carried out with the variables indicated in Table 1, achieved a percentage of variability of 74.1%.

Figure 11a shows that the variable wall covering is closely linked to the axis of dimension 1, and the other two variables (wall material and floor covering) are more closely linked to dimension 2. In particular, the variable floor covering is placed in the first quad-

rant, between the two dimensions, suggesting the nature of the materials used for the floor coverings is like those used for the wall material. These materials are characterized by a massive thickness and they are compact and consistent materials, such as marble, cement, wood. They are unlike the materials used for wall coverings, such as plaster or wallpaper, which are defined by small thicknesses. Therefore, the correlation circle can be representative of the thickness of the material components.

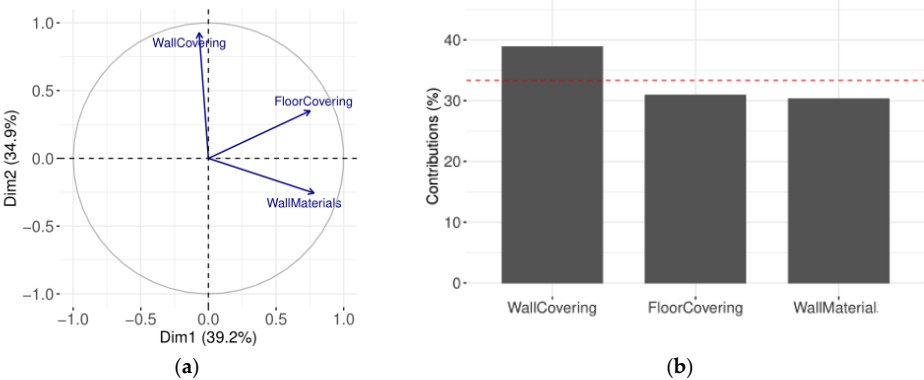

(a)                                                                          (b)

**Figure 11.** Results of the fourth PCA analysis: (**a**) the correlation circle and (**b**) percentage contributions of the factors that formed the main components (see Table 1).

The percentage contributions of the three variables in identifying the two main dimensions are shown in Figure 11b. The contribution of the wall covering is greater than that of the floor covering, which in turn is greater than that of the floor number.

In the biplot obtained (Figure 12), there is only one individual which behaves differently from the others, namely the indoor radon value of 44 Bq/m$^3$. This value is positively correlated with both components but more so with the size of the thin material components. The corresponding house, therefore, has a radon value that is more influenced by the wall covering than by the solid materials that make up the floor covering and the internal perimeter structure.

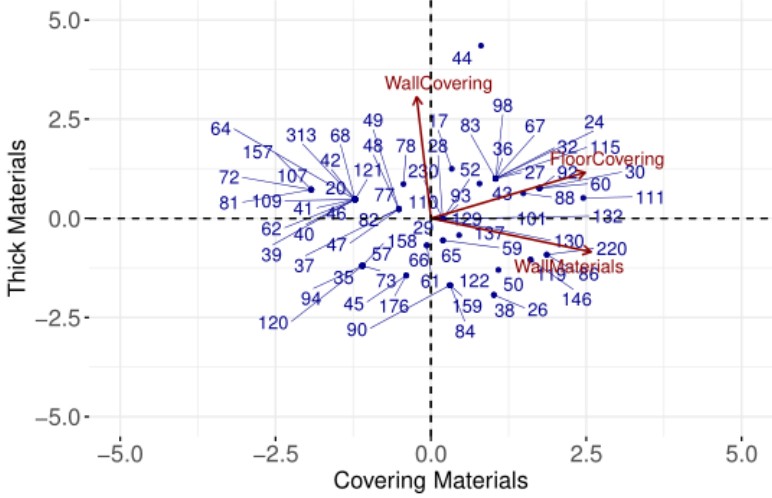

**Figure 12.** Biplot obtained from the fourth analysis (see Table 1).

In contrast to analysis number 1, the depicted biplot shows that most individuals have similar behavior around the origin of the axes of the two principal components; moreover, both components have a similar percentage value of explained variability, so the same amount of latent information is explained in both.

The 694 indoor radon data referring to dwellings located in the Campania Region were also classified through the partitive k-means algorithm. The input matrix is made up

of all the individuals, i.e., the 694 Bq/m³ data, and by columns of the variables: indoor radon, building type, floor, wall material, wall covering, floor covering, windows, doors, and postcode.

The optimal cluster number varies between k = 3 and k = 5. Applying the k-means algorithm with k = 3, k = 4 and k = 5 clusters, the highest silhouette value (0.93) corresponding to k = 5 is obtained.

The clustering groups the data according to the five Provinces of the Campania Region: Salerno, Caserta, Benevento, Avellino and Naples. Each cluster is representative of one Province (Figure 13).

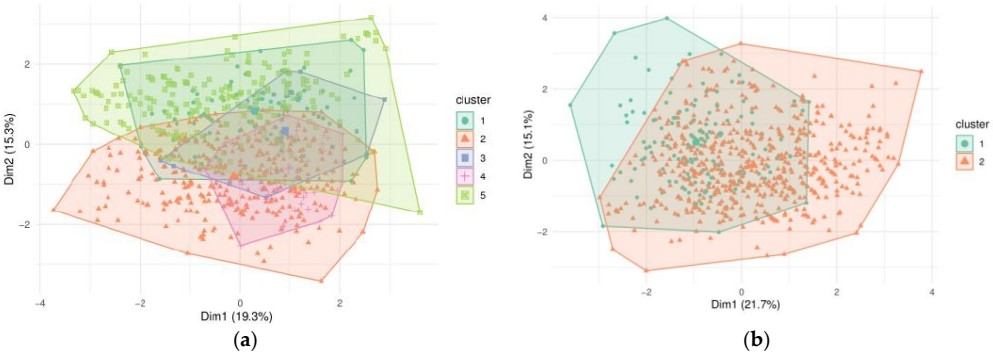

(a)    (b)

**Figure 13.** Representation of (**a**) five clusters of the first k-means analysis identified based on geographical position corresponding to the five Campania provinces (1 = Salerno, 2 = Caserta, 3 = Benevento, 4 = Avellino, 5 = Napoli) and (**b**) two clusters obtained from the second analysis.

The clustering analysis was repeated excluding the postcode variable. The highest silhouette value is for a subdivision in k = 2 clusters. The analysis of the characteristics of the two clusters shows a division between dwellings with higher and lower values of radon activity concentration (Figure 13b). The mean and standard deviation of indoor radon values in the two groups are (181, 60) Bq/m³ and (72, 28) Bq/m³, respectively.

Table 2 shows that the percentage of dwellings with walls made of tuff is higher in cluster 1 than in cluster 2, whereas the percentage of dwellings with walls made of materials such as cement and bricks, which are made of sedimentary and alluvial rocks, is higher in cluster 2 than in cluster 1. Therefore, the presence of dwellings with linoleum flooring in cluster 2 alone justifies the lower average indoor radon value in the latter.

**Table 2.** Characteristics of the two clusters identified with the latest k-means analysis.

| | Cluster 1 | Cluster 2 |
|---|---|---|
| **Radon Activity Concentration (Bq/m³)** | | |
| Mean value | 181 | 72 |
| Standard Deviation | 60 | 28 |
| **Percentage Distribution of Wall Materials (%)** | | |
| Cement | 5.6 | 8.9 |
| Bricks | 20.4 | 42.8 |
| Stones | 15.5 | 15.9 |
| Tuff | 58.5 | 25.9 |
| Other materials | 0 | 6.5 |
| **Percentage Distribution of Floor Covering Materials (%)** | | |
| Majolica | 64.1 | 60.1 |
| Wood | 0 | 2.5 |
| Linoleum | 0 | 1.3 |
| Granite | 0.7 | 0.9 |
| Moquette | 10.6 | 8.3 |
| Marble | 8.5 | 9.1 |
| Other materials | 16.1 | 17.8 |

## 4. Conclusions

Two datasets referring to building materials and dwellings in the Campania Region have been analyzed in terms of natural radioactivity through the method of Principal Component Analysis and automatic non-hierarchical partitioning.

Both the PCA and the clustering operations were performed using the RStudio software, a specific development environment for statistical data analysis, whose programming language is an open-source object-oriented language.

It was deduced that building materials of volcanic origin contribute most to the concentration of indoor radon and thoron activity. The PCA method highlighted the different influence of the most characteristic parameters.

The principal component analyses carried out on the samples of dwellings built provided a several insights: it was possible to understand how and to what extent each indoor radon value of a dwelling analyzed was influenced by the structure of the building or its geographical location, the composition of the floor or walls, or even which type of material components most influenced the interior of the dwelling in terms of radioactivity.

In conclusion, this preliminary study certainly shows that these methods of statistical analysis can also be applied for detailed and fruitful analyses of the influence of various parameters on the levels of radon and other radionuclides.

**Author Contributions:** Conceptualization, C.S. and M.L.F.; methodology, C.S., M.L.F., V.D. and G.L.V.; software, M.L.F.; validation, C.S., F.A. and M.L.F.; formal analysis, M.L.F. and F.A.; investigation, V.D. and G.L.V.; data curation, F.A., V.R., M.L.F. and V.F.; writing—original draft preparation, M.L.F. and C.S.; writing—review and editing, C.S., M.P., V.F. and A.D.; supervision, A.D., M.P., V.F. and V.R. All authors have read and agreed to the published version of the manuscript.

**Funding:** This research received no external funding.

**Institutional Review Board Statement:** Not applicable.

**Informed Consent Statement:** Not applicable.

**Data Availability Statement:** The data presented in this study are available upon request of the corresponding author. The data is not publicly available because it is subject to further analysis.

**Conflicts of Interest:** The authors declare no conflict of interest.

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
