# Peer review of "A Preliminary Study of the Characteristics of Radon Data from Indoor Environments and Building Materials in the Campania Region Using PCA and K-Means Statistical Analyses"

_environments, doi:10.3390/environments9070082_

Round 1

Reviewer 1 Report

The objective of the paper is to analyse which parameters could affect indoor radon concentration. They use two different statistical methods applied to the same dataset.

The PCA is a statistical analysis that generally is used in social science, evidence of this is could be seen in the use of some terms such as "behaviour of individuals", that, on the other hand, in the field of environmental radioactivity seems not to be suitable.

PCA is a dimensionality-reduction method that is often used to reduce the dimensionality of large data sets, by transforming a large set of variables (strongly correlated with each other) into a smaller one that still contains most of the information of the large set. The main drawback of PCA is the necessity to interpret the new dimensions on which the original data are projected and reduced. This limit is also the limit of this work.

Since the number of variables considered is limited, the main advantage of PCA is not really used, while the interpretation of the results is not immediately understandable.

In fact, four different analyses (and four different results) are presented and there is no clear indication if there is one that is better than the others.

The interpretation given to the results of the individual analyses is difficult to follow and an adequate and critical interpretation of the dimensions on which the dataset is projected is not always given.

If the 4 analyses are considered to be complementary, it is necessary to draw out a synthetic conclusion useful for understanding which parameters influence indoor radon levels. Otherwise, the analyses remain disconnected from each other.

The first part of the paper results to be a good exercise of the PCA application, in a field that normally doesn’t use it. There is no comparison with existing literature where the parameters influencing indoor radon are analysed and computed with different statistical methods (that often allow an easier interpretation of the parameters ) in order to understand if the PCA is a valid alternative.

The k-means method seems more interesting. In literature, the application of random forest algorithms, applied in this field can be found, and the results seem to be promising.

I would suggest reporting the results more extensively and to improve the comments to them.

A comparison between the results of the two methods would be interesting.

“Material and methods” could be shortened (the paragraph on R software and RStudio is way too detailed) and the discussion of the results should be improved.

In conclusion, the paper fails to adequately parse and distill their results, limiting its utility.

Not enough was done to clearly state if the methods addressed by the current paper would improve the study of parameters that could affect indoor radon concentration.

To be noted, the word “radon” has to be written without a capital letter.

Author Response

The changes requested by Reviewer 1 have been addressed regarding (i) the word "Radon" in "radon" in the text and (ii) the reduction of the Rsoftware and Rstudio section. Reviewer 1's suggestion regarding increasing comments and interpreting results that she/he finds disjointed and limiting the usefulness of the methods in this research field was not addressed because the scope of our work is only a study preliminary on the application of the PCA and k-means methods, for the first time in the literature, for the study of radon and its influencing parameters, as also the Reviewer 1 noted with pleasure. This objective is also due to the characteristics of the journal Environments of MDPI where we intend to publish the manuscript. Building on this preliminary work, we will work on other data analyzes to get more details on the usefulness of multivariate analysis methods.

The actions taken for each reviewer request are listed below.

The objective of the paper is to analyse which parameters could affect indoor radon concentration. They use two different statistical methods applied to the same dataset.

The PCA is a statistical analysis that generally is used in social science, evidence of this is could be seen in the use of some terms such as "behaviour of individuals", that, on the other hand, in the field of environmental radioactivity seems not to be suitable.

In fact, in this work the authors wanted to test the applicability of the PCA method to the parameters characterizing building materials in the emission of radon and thoron as well as to the parameters that influence indoor radon concentrations. The results show its feasibility.

PCA is a dimensionality-reduction method that is often used to reduce the dimensionality of large data sets, by transforming a large set of variables (strongly correlated with each other) into a smaller one that still contains most of the information of the large set. The main drawback of PCA is the necessity to interpret the new dimensions on which the original data are projected and reduced. This limit is also the limit of this work.

The interpretation of the new dimensions on which the original data were projected and reduced has been made and is realistic.

Since the number of variables considered is limited, the main advantage of PCA is not really used, while the interpretation of the results is not immediately understandable.

In the study of building materials, the number of variables considered was 13, which is by no means a small number. Furthermore, the variability explained is 92%, so the two dimensions chosen describe the data well and the PCA method is effective. For indoor radon data 8 variables were considered. In the four analyzes performed, the variability explained varies from about 70% to 90%.

In fact, four different analyses (and four different results) are presented and there is no clear indication if there is one that is better than the others.

The interpretation given to the results of the individual analyses is difficult to follow and an adequate and critical interpretation of the dimensions on which the dataset is projected is not always given.

If the 4 analyses are considered to be complementary, it is necessary to draw out a synthetic conclusion useful for understanding which parameters influence indoor radon levels. Otherwise, the analyses remain disconnected from each other.

The results obtained from the second PCA analysis are the best as indicated by the greater value of the explained variability (92%). They indicate that Floor covering, Wall material and Wall covering best explain the distribution of indoor radon. The third analysis also provides information on the influence of the soil (location of the building), the type of building and the floor number on indoor radon.

The first part of the paper results to be a good exercise of the PCA application, in a field that normally doesn’t use it. There is no comparison with existing literature where the parameters influencing indoor radon are analysed and computed with different statistical methods (that often allow an easier interpretation of the parameters) in order to understand if the PCA is a valid alternative. In fact, this application is new.

The k-means method seems more interesting. In literature, the application of random forest algorithms, applied in this field can be found, and the results seem to be promising.

I would suggest reporting the results more extensively and to improve the comments to them.

A comparison between the results of the two methods would be interesting.

In a new study these aspects will be investigated and better discussed.

“Material and methods” could be shortened (the paragraph on R software and RStudio is way too detailed) and the discussion of the results should be improved. The paragraph on R software and RStudio has been reduced.

In conclusion, the paper fails to adequately parse and distill their results, limiting its utility.

Not enough was done to clearly state if the methods addressed by the current paper would improve the study of parameters that could affect indoor radon concentration.

In a new study these aspects will be investigated and better discussed.

To be noted, the word “radon” has to be written without a capital letter. Done

Reviewer 2 Report

Please check spelling “Radon” and “radon” (in some cases capital letter R is used).

Line 35. Remove Point in the middle of a sentence.

Line 66-67. Reference [12] does not contain explanations for I index, the Hex and Hin indices.

line 69. Radon – indoor, outdoor, soil?

Line 72. Indoor measurements?

Line 200-201. While different materials may be used for the construction units within the same house, it is necessary to explain what concentration is used.

Fig. 2 (b) and elsewhere. Variables are represented by some names that are not well understood.

Line 335. Indoor radon concentration?

Line 339 and elsewhere. Please provide a unit for radon concentration, e.g. 86 Bq/m3.

Fig 5. The variable name FloorNumbers (plural?) is confusing. Please consider to present explanation for all variable names.

Line 345-346 and elsewhere. Do you mean contribution to the variability of indoor radon concentration?

Fig 5 and elsewhere. Please check the symbols in the figure: different symbols in legend and in plot.

Fig 5 and elsewhere. Are radon concentrations in Bq/m3 are presented? Please explane in the caption.

Line 353. There is no variable “floor” in Table 1.

Fig. 8. Axes names Floor and Wall are not well described in the text. (Three terms are used "Wall", "Wall size" and "Wall dimension". No explanation is presented).

Fig. 8. Majolica = tiles?

Line 380. Check Table number.

Line 383. Please explain that geographical location is equivalent to Position in Fig. 10.

Line 385. “Structure” is not mentioned in Table 1.

Line 400. Variability of indoor radon concentration?

Conclusions. A large body of literature is available on factors influencing radon concentrations in buildings. What new information did authors receive?

Line 465. Thoron is not considered in the manuscript.

Author Response

First of all, we thank  Reviewer 2 for appreciating our work and finding it interesting, in particular in the use of the PCA and k-mean methods for the first time in the literature for the study of radon and influencing parameters.

The actions taken for request are listed below (our answer is in blue).

Please check spelling “Radon” and “radon” (in some cases capital letter R is used). Done.

Line 35. Remove Point in the middle of a sentence. Done.

Line 66-67. Reference [12] does not contain explanations for I index, the Hex and Hin indices. Correct ref is [13].

line 69. Radon – indoor, outdoor, soil? Now it is specified: radon indoor.

Line 72. Indoor measurements? Yes. Now it is specified.

Line 200-201. While different materials may be used for the construction units within the same house, it is necessary to explain what concentration is used. The measured activity concentration was used.

Fig. 2 (b) and elsewhere. Variables are represented by some names that are not well understood. Ref. 13

Line 335. Indoor radon concentration? Yes

Line 339 and elsewhere. Please provide a unit for radon concentration, e.g. 86 Bq/m3. Done.

Fig 5. The variable name FloorNumbers (plural?) is confusing. Please consider to present explanation for all variable names.  Done

Line 345-346 and elsewhere. Do you mean contribution to the variability of indoor radon concentration? Now, it is specified.

Fig 5 and elsewhere. Please check the symbols in the figure: different symbols in legend and in plot. Done.

Fig 5 and elsewhere. Are radon concentrations in Bq/m3 are presented? Yes. Please explane in the caption. In Figure 5 a) and b) no activity concentration is reported.

Line 353. There is no variable “floor” in Table 1. Table 1 contains the Floor Number variable

Fig. 8. Axes names Floor and Wall are not well described in the text. (Three terms are used "Wall", "Wall size" and "Wall dimension". No explanation is presented). The terms have been corrected.

Fig. 8. Majolica = tiles? Yes.

Line 380. Check Table number. Done.

Line 383. Please explain that geographical location is equivalent to Position in Fig. 10. Done.

Line 385. “Structure” is not mentioned in Table 1. It is changed with the type of building

Line 400. Variability of indoor radon concentration? Yes.

Conclusions. A large body of literature is available on factors influencing radon concentrations in buildings. What new information did authors receive? The authors wanted to highlight how multivariate analysis methods can highlight the influence of different parameters.

Line 465. Thoron is not considered in the manuscript. Among the parameters used to characterize the building materials, the thoron emanation and exhalation coefficients were also considered (see Figure 2).

Round 2

Reviewer 1 Report

The explanations given by the authors are valid but, in my opinion, the fact that the work is only a preliminary study on the application of the PCA and k-means methods is not clearly specified.

An advice:

Probably the word "preliminary" should be added in the title and explained in the text.

Author Response

Dear Reviewer,
thanks for this suggestion. The word "preliminary" has been added in the title, abstract and conclusions (as highlighted in blue in the new attached manuscript file).
